# High-Intensity Interval Training Improves Glycemic Control, Cellular Apoptosis, and Oxidative Stress of Type 2 Diabetic Patients

**DOI:** 10.3390/medicina59071320

**Published:** 2023-07-17

**Authors:** Hadeel A. Al-Rawaf, Sami A. Gabr, Amir Iqbal, Ahmad H. Alghadir

**Affiliations:** 1Department of Clinical Laboratory Sciences, College of Applied Medical Sciences, King Saud University, Riyadh 11433, Saudi Arabia; hadeelar12345@gmail.com; 2Department of Rehabilitation Sciences, College of Applied Medical Sciences, King Saud University, Riyadh 11433, Saudi Arabia; dr.samigabr@gmail.com (S.A.G.); aalghadir@hotmail.com (A.H.A.)

**Keywords:** mitochondria DNA (mt-DNA), p53, cytochrome c oxidase, oxidative stress, diabetes, high-intensity interval training (HIIT)

## Abstract

*Background and Objectives*: Physical exercise is an important therapeutic modality for treating and managing diabetes. High-intensity interval training (HIIT) is considered one of the best non-drug strategies for preventing and treating type 2 diabetes mellitus (T2DM) by improving mitochondrial biogenesis and function. This study aimed to determine the effects of 12 weeks of HIIT training on the expression of tumor suppressor protein-p53, mitochondrial cytochrome c oxidase (COX), and oxidative stress in patients with T2DM. *Methods:* A total of thirty male sedentary patients aged (45–60 years) were diagnosed with established T2DM for more than five years. Twenty healthy volunteers, age- and sex-matched, were included in this study. Both patients and control subjects participated in the HIIT program for 12 weeks. Glycemic control variables including p53 (U/mL), COX (ng/mL), total antioxidant capacity (TAC, nmole/µL), 8-hydroxy-2′-deoxyguanosine (8-OHdG, ng/mL), as well as genomic and mitochondrial DNA content were measured in both the serum and muscle tissues of control and patient groups following exercise training. *Results:* There were significant improvements in fasting glucose levels. HbA1c (%), HOMA-IR (mUmmol/L^2^), fasting insulin (µU/mL), and C-peptide (ng/mL) were reported in T2DM and healthy controls. A significant decrease was also observed in p53 protein levels. COX, 8-OhdG, and an increase in the level of TAC were reported in T2DM following 12 weeks of HIIT exercise. Before and after exercise, p53; COX, mt-DNA content, TAC, and 8-OhdG showed an association with diabetic control parameters such as fasting glucose (FG), glycated hemoglobin (HbA1C, %), C-peptide, fasting insulin (FI), and homeostatic model assessment for insulin resistance (HOMA-IR) in patients with T2DM. These findings support the positive impact of HIIT exercise in improving regulation of mitochondrial biogenesis and subsequent control of diabetes through anti-apoptotic and anti-oxidative pathways. *Conclusions:* A 12-week HIIT program significantly improves diabetes by reducing insulin resistance; regulating mitochondrial biogenesis; and decreasing oxidative stress capacity among patients and healthy controls. Also; p53 protein expression; COX; 8-OhdG; and TAC and mt-DNA content were shown to be associated with T2DM before and after exercise training.

## 1. Introduction

Physical activity and exercise offer a wide range of health benefits. Physical activity was reported for all ages to minimize cardiovascular risks, improve immune function, energy balance, psychological status, and improve overall health for all ages [1,2].

Type 2 diabetes mellitus is considered one of the most serious chronic diseases that often follows obesity. It is characterized by lower cardiorespiratory fitness, a major predictor of mortality, and severe complications in patients with diabetes [3,4]. Multiple potential factors, such as sedentary lifestyle, eating habits, physical inactivity, and obesity have been strongly associated with higher incidence rates of diabetes worldwide. These factors impose a substantial burden on healthcare systems and contribute to more severe complications among diabetic patients, which necessitates new strategies for disease management or treatment [5,6,7].

In most diabetic cases, complications such as those affecting microvascular and macrovascular tissues, and subsequently the pathogenesis of diabetes develop silently through genetic or other factors affecting glycemic control. Specifically, in cases like type 2 diabetes mellitus, the progression often proceeds with undiagnosed symptoms for several years [8,9].

Mitochondria in human tissues are considered one of the organelles most responsible for the aerobic synthesis of molecular ATP, which supplies the required chemical energy for vital cellular processes [10,11]. Previously, it was reported that the number of mitochondria and their function was significantly reduced in tissues associated with diabetic and metabolic diseases [12,13]. In patients with diabetes, synthesis of ATP was significantly reduced or failed in response to insulin infusion owing to the reduction in the number of mitochondria in skeletal muscle [14,15]. Skeletal muscle was shown to be sensitive and adaptive to many external stimuli, especially exercise training programs. These adaptations significantly depend mainly on frequency, intensity, duration, and exercise mode. Mitochondrial biogenesis is one of the most important adaptive responses to aerobic exercise training, which increases the expression of mitochondria mRNA, number, protein synthesis, and oxidative activity [16,17,18,19,20].

Oxidative stress and increased production of reactive oxygen species (ROS) were shown to be linked with mitochondrial dysfunction in type 2 diabetes mellitus (T2DM), whereas impaired capacity for glycogen and lipid oxidation and insulin resistance of peripheral tissue was significantly reported in T2DM patients [21,22]. Previously, it was reported that the pathogenic mechanism underlying insulin resistance in skeletal muscle of T2DM was an increment in tissue oxidative stress [23,24,25,26], which ultimately leads to reduction in glucose and fatty acid uptake [22,27,28].

P53 is a well-known tumor suppressor protein that is critical in regulating cell growth and preventing cancer development. In addition to its role in tumor suppression, p53 also plays a key role in regulating apoptosis or programmed cell death. When cells are damaged or undergo stress, p53 is activated to initiate apoptosis and remove the damaged cells from the body [29,30,31,32].

Mitochondrial cytochrome c, on the other hand, is a protein located in the inner membrane of mitochondria, the organelles responsible for generating energy in cells. In healthy cells, cytochrome c is sequestered within the mitochondria, but when activating stress, it is released into the cytoplasm, which triggers the apoptotic pathway by activating caspase enzymes [29,30,31,32]. Studies have shown that p53 and mitochondrial cytochrome c function together regulate apoptosis and maintain cellular homeostasis. In particular, p53 has been shown to induce cytochrome c release from mitochondria in response to DNA damage, activating the apoptotic pathway. Additionally, p53 has been shown to regulate the expression of Bcl-2 family proteins, which play a key role in regulating mitochondrial membrane permeability and cytochrome c release [29,30,31,32]. The dysregulation of p53 and cytochrome c have been implicated in developing and progressing various diseases, including cancer, neurodegenerative disorders, and cardiovascular disease [29,30,31,32].

Moreover, tumor suppressor p53, which functions as a crucial transcription factor, has been shown to play a direct role in energy metabolism by balancing aerobic respiration with glycolysis in the cell [33,34]. Studies have also reported that p53 is responsible for the decline in mitochondrial respiration via mutation in cytochrome oxidase deficient homolog 2 (SCO2) genes, which suppress assembly of COX (cytochrome c oxidase), a critical component and the major site of oxygen utilization in the respiratory chain, resulting in aerobic respiratory failure [35,36]. P53 was shown to improve aerobic exercise capacity and augment skeletal muscle mitochondrial DNA content, suggesting that p53 may also increase mitochondrial oxidation in skeletal muscle by regulating mitochondrial biogenesis [37].

Exercise training of varying modes is considered one of the best non-drug strategies in the prevention and treatment of T2DM. It showed to play a part in the increment of muscle oxidative capacity, mitochondrial function, and insulin sensitivity in humans [38,39,40,41,42,43]. In metabolically active tissues, physical activity resulting from exercise is one of the best-known strategies to improve mitochondrial function and human health [44,45]. The involvement of p53 in regulating mitochondrial DNA content, respiration, function and oxidative stress, and skeletal muscle insulin resistance following regular exercise is fully elucidated [46,47,48].

In middle-aged (45–60 years) and older individuals, type 2 diabetes is more prevalent, with the incidence of the disease increasing with age [49,50,51,52]. Overall, selecting the age range of 45–60 years allows for a more targeted investigation of the effects of exercise on biological biomarkers in a population that is representative of those most affected by type 2 diabetes and its complications [49,50,51,52]. Therefore, it was proposed in this study that studying the effects of exercise on some biomarkers in subjects aged 45–60 years can provide important insights into the potential benefits of exercise in older individuals with type 2 diabetes.

However, in middle-aged type 2 diabetes, little is known about the effects of high-intensity interval training (HIIT) on the p53 regulation mechanism and mitochondrial function in patients with T2DM. Therefore, this study aimed to determine the effects of a 12-week HIIT program on p53 expression, mitochondrial cytochrome c, and oxidative stress in patients with T2DM.

## 2. Materials and Methods

### 2.1. Subjects

A total of thirty male sedentary patients aged (45–60 years) diagnosed with established T2DM for more than five years were invited to participate in this study. In addition, twenty healthy subjects from a population undergoing a standard annual physical examination and biological measurements for medical insurance were selected as controls. Patients with obesity (BMI) ≥ 35 kg/m^2^), lower fasting blood glucose (<140 mg/dL), and use of drugs or exogenous insulin, which may affect the data, were excluded from this study. Patients with chronic diseases such as kidney and liver diseases or severe diabetic complications such as neuropathy, retinopathy, neuromuscular, cardiopulmonary, and physical disability or movement limitations were also excluded from this study. All participants were sedentary with little or no physical activity during daily routine activities such as work and transportation. The participants were instructed not to change their normal eating habits during the entire data collection period. The study was performed between June and December 2013. The study protocol conformed to the ethical guidelines of the 1975 Declaration of Helsinki. It was reviewed and approved by the Ethics Sub-Committee of King Saud University, Kingdom of Saudi Arabia, under file number ID: RRC-2013-025. Demographic and clinical data of the participants are shown in Table 1.

### 2.2. Exercise Training Program

The high-intensity interval training (HIIT) program was entirely performed in the exercise labs at King Saud University, Kingdom of Saudi Arabia, under the supervision of specialist physiotherapists. Patients and control subjects participated in the HIIT program for 40 min/3 sessions/week for 12 weeks by using an electronic treadmill (Vegamax, made in Taiwan). The HIIT program consisted of 4 × 4 min intervals at 80–85% of HR max, with 3-min active recovery at 70% of HR max between intervals. The participants started a warm-up for 10 min at 50% of maximal heart rate (HR max) and 5 min cool-down before initial HIIT program sessions, as previously reported [44]. In this study, the HIIT exercise program was designed to perform physical activities corresponding to 30–45% of VO_2_ max uptake for each participant [45]. To reach for improved fitness, each participant’s heart rate and the Borg scale of perceived exertion were checked to maintain exercise intensity and avoid training adaptations throughout the exercise period.

### 2.3. Skeletal Muscle Biopsy and Blood Samples

Both muscle biopsy and blood samples were collected from all participants before and after the HIIT exercise program. Following an overnight fast, serum samples were extracted using centrifugation for 10 min of the blood samples. In addition, all biopsies were collected in a fasted state early morning at baseline one week before exercise intervention, to allow the biopsy wound to heal and any inflammation from the biopsy itself to subside before another biopsy following 14 h of rest after the last HIIT session, as previously reported [53,54]. In this technique, a licensed medical general practitioner (GP) with experience in percutaneous needle biopsies took the muscle biopsies for the study. Biopsies were taken from alternating legs in randomized order between subjects from vastus lateralis. First, local anesthesia (2% Carbocain, AstraZeneca, Sodertalje, Sweden) was injected at the biopsy site. A small incision was made, and approximately 150 mg of wet tissue was removed with a Weil–Blakesley conchotome or a 5 mm Bergstrom needle with a manually applied suction. Finally, the GP instructed the subjects to take the Band-Aid off 1 to 2 h post-biopsy and naturally allow the plasters to remove over time. If there were any abnormalities with healing or severe pain post-biopsy, the subjects were instructed to communicate with the investigator or the GP, who helped accordingly.

Both serum and muscle samples were immediately frozen under liquid nitrogen and stored at −80 °C until further analysis of p53 protein, cytochrome c, 8-OHdG, TAC, genomic, and mitochondrial DNA content [55].

### 2.4. Quantification of Mitochondrial DNA Content by Real-Time PCR

Copy numbers of both genomic and mitochondrial DNA per milligram wet tissue weight were estimated from muscle samples of all participants at baseline and following the HIIT program for 12 weeks. Total DNA was precipitated from approximately 15 mg of frozen tissue and concentrations were determined as previously described [56,57]. In this test, total DNA was precipitated from adipose tissue samples (~10 mg) homogenized in DNAzol (Molecular Research Center, Cincinnati, OH, USA). The DNA was dissolved in 100 μL Tris-EDTA pH 8. Five microliters of a 50 times DNA dilution was used for PCR amplification with QuantiTect SYBR Green PCR Master Mix (Qiagen, Hilden, Germany) containing 2.5 pmol of each primer in a total volume of 25 μL. Levels of mtDNA were determined by real-time PCR using an MX3005P QPCR machine (Stratagene, La Jolla, CA, USA), and the concentration of mtDNA per milligram of tissue was used as an estimate of the amount of mitochondria per milligram of tissue [56,57].

### 2.5. Estimation of Glycemic Control Parameters

Pre- and post-HIIT training tests, serum glucose, and c peptide levels were measured in the routine analysis laboratory using spectrophotometric and chemiluminescent methods. HPLC determined hbA1c levels in erythrocytes in the same laboratory. Serum levels of insulin were estimated in all subjects using an ELISA kit (Insulin ELISA kit human, KAQ1251, Invitrogen Corporation, Camarillo, CA, USA). By using a pre-validated homeostasis model assessment of insulin resistance (HOMA-IR), insulin resistance was evaluated in the fasting state. The results of IR were significantly calculated in the fasting insulin (IF) and fasting glucose (GF) as follows: HOMA-IRZ (IF × GF)/22.5 [48,49].

### 2.6. Estimation of Serum Cytochrome c (COX) and p53

Cytochrome c and p53 were measured in both serum and muscle samples with competitive ELISA kits obtained from Chemicon International, Temecula, CA, USA, and Bender MedSystems, Burlingame, CA, USA, respectively. The procedures of both p53 and cytochrome c ELISA kits were performed according to the manufacturer’s instructions [50].

### 2.7. Estimation of Oxidative Stress and Antioxidant Capacity

Total antioxidant capacity (TAC) was measured in serum and muscle samples by Colorimetric Assay Kit (Catalog #K274-100; BioVision Incorporated, Milpitas, CA, USA). The antioxidant equivalent concentrations were measured at 570 nm as a function of Trolox concentration according to the manufacturer’s instructions: Sa/Sv = nmol/µL or mM Trolox equivalent (where Sa is the sample amount (in nmol) read from the standard curve; Sv is the undiluted sample volume added to the wells). Serum 8-OHdG as a marker of DNA damage was also estimated in all subjects by using a commercially available ELISA kit (DNA damage ELISA Kit, Product #: ADI-EKS-350, Enso life sciences Co., Farmingdale, NY, USA) [51].

### 2.8. Sample Calculations

A sample comprising 50 subjects was included in this study. G ∗ Power program for Windows (version 3.1.9.7, Heinrich-Heine-University) was used to measure the power of the sample size of 50 subjects. Using the *t*-test with a significance level of 0.05, the total sample of 50 achieves a power of 95% with an effect size of 0.92, Df = 45.74, critical *t* = 1.68, and noncentrality − α = 3.34. Based on the incidence of diabetes, participants were then divided into two groups: a healthy group (*n* = 20) and a T2DM group (*n* = 30).

### 2.9. Statistical Analysis

Data were expressed as mean ± standard deviation (SD). Repeated measures ANOVA were performed to evaluate the changes in COX, p53, glycemic control, and oxidative stress variables pre- and post- HIIT exercise training within and between groups (pre- and post-study). Statistical analysis was performed using SPSS software (version 16.0). All data levels set at *p* < 0.05 were reported as significant (SPSS statistical 13.0 software version for Windows; SPSS Inc., Chicago, IL, USA).

## 3. Results

To study the potential effects of T2DM on muscular mitochondria, p53, COX, and mitochondrial DNA content as well as oxidative stress parameters, TAC, and 8-OHdG were estimated as biomarkers of mitochondrial changes in serum and muscle tissues of patients with T2DM and control subjects (Figure 1). Baseline data showed a significant increase in the levels of p53, COX, and 8-OHdG (Figure 1A,C), with a decrease in TAC and mt-DNA content (Figure 1B,D,E) in serum and muscle tissues of diabetic patients (*p* = 0.001) compared to healthy control subjects.

In this study, the effects of HIIT exercise training for 12 weeks on T2DM and its other related complications were evaluated both in healthy subjects (*n* = 20) and T2DM (*n* = 30) patients. Thus, adiposity markers (BMI, waist, Hips, and WHR) and glycemic control parameters (FG, serum C-peptide, FI, HbA1c (%), and IR) were estimated in serum samples of all subjects pre- and post-HIIT exercise training, as shown in Table 1.

Compared to pre-exercise data, VO_2_ max values as measures of physical fitness showed an improvement with a significant decrease in adiposity markers in healthy control subjects (CON; *p* = 0.01) and T2DM patients (T2DM; *p* = 0.001) following 12 weeks of HIIT exercise training (Table 1). In addition, significant improvement in diabetic control parameters was reported in healthy subjects (CON; *p* = 0.01) and type 2 DM patients (T2DM; *p* = 0.001) after HIIT exercise training, with a significant decrease in FG, IR, HbA1c (%), and an increase in the levels of serum FI and c-peptides (Table 1).

The potential effects of HIIT exercise training on p53 protein, COX, TAC, 8-OHdG, and mt-DNA content were evaluated in all participants following exercise training. Post-exercise data showed a significant reduction in the expression levels of p53, Cox, and 8-OHdG, with a significant increase in the levels of mt-DNA and TAC in serum and muscle tissues of control (Figure 2 and Figure 3) and diabetic cases (Figure 4 and Figure 5) following HIIT exercise training for 12 weeks.

## 4. Discussion

The present study reported significant improvement in adiposity markers: BMI and waist-to-hip ratio (WHR) reduced in both control and T2DM patients following 12 weeks of the HIIT program. In diabetic and control subjects, glycemic control parameters such as FG, HbA1c (%), HOMA-IR, fasting insulin, and C-peptide showed significant improvements toward normal control values following exercise training. HIIT training at moderate interval training was reported to be beneficial for patients with stable post-infarction heart failure who were undergoing optimal medical treatment, including β-blockers and angiotensin-converting enzyme inhibitors. In that study, patients who underwent interval training reported a significant improvement in brachial artery flow-mediated dilation (endothelial function), while mitochondrial function in lateral vastus muscle increased with aerobic interval training alone, compared to those who underwent moderate continuous training. Therefore, it was concluded that exercise intensity with interval training was an important factor in improving aerobic capacity, endothelial function, and quality of life in patients [58,59,60]. Moreover, applying HIIT training at different intervals (lower, moderate, and high activity) with intermittent bursts of strenuous activity, was shown to be the most suitable for different physiological adaptations in many diseases, such as diabetes [61]. Previously, beneficial effects of regular exercise were reported in diabetic patients with or without complications and/or only insulin resistance [62,63]. It was reported that HIIT exercises for 8 weeks were shown to improve glycemic control and pancreatic β Cell function, preserve insulin secretion, and enhance insulin sensitivity of type 2 diabetes [64,65]. Additionally, interval training exercise was shown to improve free-living and postprandial glycemic control parameters of diabetic patients when compared to continuous exercise with matched time duration and oxygen consumption [66].

In addition, HIIT exercise training for 8 weeks was found to significantly reduce total body fat and visceral adiposity, which confirms our results regarding improvements in adiposity parameters such as BMI and WHR. Moreover, these results concluded that HIIT training was found to be superior in reducing more visceral adiposity levels compared with conventionally applied exercise training programs [67]. It was proposed recently that HIIT training programs with different intensities enhance glycemic control of diabetic cases though improved adipose tissue sensitivity of liver insulin, which subsequently leads to a potential increase in insulin [67,68,69]. In patients with diabetes, the improvements in diabetic status depend on the potential adaptation of the patient’s skeletal muscle and sensitivity to frequency, intensity, duration, and mode of exercise performed [11,12].

In most cases, diabetic patients are characterized by a failure or reduction in the synthesis of ATP, which may be related to insulin infusion and a lower number of mitochondria in skeletal muscle [14,15,16,17,18]. Mitochondria constitutes the most important resource for utilizing molecular ATP needed for vital cellular processes in human tissues [11,12,13]. Therefore, any change in the physiological function of mitochondria significantly affects diabetic patients.

This study estimated the p53, mitochondrial COX, and mt-DNA content levels in serum and muscle tissues of T2DM patients and healthy control participants before and after HIIT exercise training for 12 weeks. In patients with T2DM, baseline values showed higher levels of p53, COX, and lower mt-DNA content in the serum and muscle tissues of patients compared with healthy control subjects. However, a significant decrease in the levels of p53 and cytochrome c and an increase in mt-DNA levels were reported in both patients and control subjects following 12 weeks of HIIT exercise. The data obtained may suggest the role of mitochondria in improving diabetes after exercise by reducing the expression of p53 protein and COX.

Mitochondrial biogenesis is one of the most important adaptive responses to aerobic exercise training, which increases the expression of mitochondria mRNA, number, protein synthesis, and oxidative activity [19,20,21,22,23,24,25,26,27,28,29,30,31,32]. Previously, it was reported that elevated glucose concentrations induce significant apoptosis in human β–cells due to constitutive expression of the Fas ligand [70]. Moreover, cytochrome c was released from mitochondria as a result of the interaction that occurs between apoptotic signals and cell surface receptors. With the help of some cytosolic proteins [71], the released cytochrome c in the cytoplasm of apoptotic cells assists caspases enzymes in activating the degradation of cellular DNA by the action of endonucleases [72]. Consistent with our results, previous studies showed that increased glucose levels induce apoptosis by expression of p53 protein and activation of cytochrome c-activated caspase-3 in response to DNA damage and subsequent decrease in mt-DNA content [73,74].

Previous studies reported that p53 might have a protective role in pancreatic β-cell death and that the increased expression of p53 was significantly associated with apoptotic death of pancreatic β–cells [75]. It was further reported that p53 is responsible for a decline in mitochondrial respiration via mutation in cytochrome oxidase deficient homolog 2 (SCO2) genes, which suppress the assembly of COX (cytochrome c oxidase), a critical component and the major site of oxygen utilization on the respiratory chain, resulting in aerobic respiratory failure [35,36]. P53 was also shown to improve aerobic exercise capacity and augment skeletal muscle mitochondrial DNA content, suggesting that p53 may also increase mitochondrial oxidation in skeletal muscle by regulating mitochondrial biogenesis [37].

Regarding the effects of the HIIT program on mitochondrial mt-DNA, it was shown that the increase of mt-DNA content in muscle tissue of diabetic and control subjects might be related to the regulatory role of p53 in modulating mitochondrial content and exercise performance [76,77,78]. In exercise models with acute activity, posttranslational modifications and subcellular localization of p53 to mitochondria facilitate its role in controlling contractile-induced mitochondrial biogenesis [78,79,80,81]. P53 is clearly involved in regulating mitochondrial biogenesis, cellular metabolism in the cell, and regulation of mt-DNA copy number and transcriptional activity [82,83,84,85]. In experimental models, exercise training decreased the level and expression of p53 in skeletal muscles with a subsequent increase in mitochondrial DNA content in skeletal muscle [82,86].

The suppression of p53 protein expression by HIIT exercise may protect myofibers from apoptosis, promote cell survival, and relieve insulin resistance and metabolic abnormalities associated with DM, providing a qualitatively different mode of p53 function as an antioxidant defense in cells [83,87]. The decrease in cytochrome c levels following exercise may be due to the regulation of p53 to ATP-generating pathways such as mitochondrial biogenesis [88,89], which catalyze the transfer of electrons from reduced cytochrome c to molecular oxygen. The loss of p53 results in decreased oxygen consumption and aerobic respiration and promotes a switch to glycolysis, thereby reducing endurance during physical exercise [24,90,91].

It was previously reported that oxidative phosphorylation systems require the expression and maintenance of nuclear and mitochondrial genomes [92,93]. The expression rates of mitochondrial DNA (mt-DNA) genomes should be in a regular state to avoid many pathological syndromes [90,91,92]. Any change in the mt-DNA expression of human skeletal muscle by point mutations, deletion, duplication, and depletion of mt-DNA can severely affect its functionality [93,94,95].

High-intensity interval training (HIIT) has improved mitochondrial function in individuals with type 2 diabetes, including changes in mitochondrial parameters such as p53, COX, and mt-DNA content [96,97,98,99,100]. HIIT has been shown to increase p53 expression in skeletal muscle, which may help to protect against oxidative stress by activating antioxidant pathways and promoting mitochondrial biogenesis. Moreover, HIIT has been shown to increase COX activity and expression in skeletal muscle, as well as increasing of mt-DNA content in skeletal muscle, which may improve mitochondrial function and enhance cellular energy production. These changes may also have implications for cellular apoptosis. Mitochondria play a key role in regulating cellular apoptosis, and dysfunction of mitochondrial pathways can contribute to developing chronic diseases, including type 2 diabetes. HIIT may help to reduce the risk of cellular apoptosis and prevent the development of chronic diseases, especially in patients with T2DM mellitus, by improving mitochondrial function and reducing oxidative stress [96,97,98,99,100]. Thus, when implementing HIIT exercise programs for diabetes management, the benefits of improved glucose control, cardiovascular fitness, and time efficiency may outweigh the potential risks for many individuals. In addition, HIIT can be performed using a variety of exercises and can be adapted to different fitness levels and physical limitations [97,98,99,100].

To improve the exercise capacity in patients with type 2 diabetes, non-drug therapies were postulated to regulate oxidative stress and maintain mitochondrial function [101]. Thus, the effect of oxidative stress on mitochondria was considered in this study. TAC and 8-OHdG were estimated in serum and muscle tissues of control and diabetic patients’ pre- and post-HIIT program. At baseline, the data showed a significant increase in the levels of 8-OHdG and a decrease in the levels of TAC activity. The data were closely correlated with a reduction in mt-DNA content in muscle tissue samples. When the patients and control group participated in the HIIT program for 12 weeks, improvements in the levels of TAC and lower levels of 8-OHdG oxidative markers of DNA damage were significantly reported in control and T2DM patients. The data significantly correlated with increased mt-DNA content in muscular tissues.

Metabolic and cardiovascular disorders associated with diabetes were shown to be associated with the initiation of free radical oxidative stress and excess ROS production, which in turn exert significant mitochondrial dysfunction in skeletal muscle and overall cardiovascular complications [102,103]. Consistent with our results, it was also reported that mitochondrial dysfunction in tissues might be due to proinflammatory responses and disturbance in redox homeostasis, which ultimately produces severe cellular damage with an increment in insulin resistance [79,81,100,101,102]. Previously, it was reported that increased mitochondrial DNA content and function were significantly associated with reduced oxidative stress and improved insulin resistance attenuation by exercise training [98,99,100,101,102,103].

Finally, the data of this study were aligned with others who reported that p53 improves aerobic exercise capacity and augments skeletal muscle mitochondrial DNA content [95,96], suggesting that p53 may also increase mitochondrial oxidation in skeletal muscle by regulating mitochondrial biogenesis and decrease in oxidative stress capacity [82].

### 4.1. Strength

This study has investigated the effects of high-intensity interval training (HIIT) on various biomarkers related to apoptosis, oxidative stress, and mitochondrial DNA in individuals with type 2 diabetes, including middle-aged. There is growing evidence to suggest that high-intensity interval training (HIIT) can improve cellular apoptosis and oxidative stress biomarkers in individuals with diabetes. The study provides some potential strengths using biomarkers P53, COX, mitochondrial DNA, TAC and 8-OHdG to assess the effects of HIIT on type 2 diabetes:(a)Scientific contribution: By inducing cellular stress through high-intensity exercise bouts, HIIT may activate cellular signaling pathways that improve mitochondrial function, increase antioxidant capacity, and reduce inflammation.(b)Objective measurement: Measuring biomarkers such as apoptosis (P53, COX), oxidative stress (TAC and 8-OHdG), and mitochondrial DNA can provide an objective measurement of the effects of HIIT on cellular and molecular processes related to type 2 diabetes.(c)Mechanistic insights: Measuring biomarkers can help to elucidate the mechanisms through which HIIT improves glucose control and insulin sensitivity in individuals with type 2 diabetes.(d)Personalized medicine: Biomarkers can potentially be used to identify individuals who are most likely to benefit from HIIT and monitor the effects of HIIT individually.

### 4.2. Limitations

The selected sample size considerably explained the changes in mitochondrial parameters observed following HIIT training and suggested that this exercise intervention may be an effective strategy for improving mitochondrial function and reducing the risk of chronic diseases, including type 2 diabetes. However, further research based on larger cohort samples of patients with T2DM mellitus is required to fully understand the mechanisms underlying these changes and their long-term implications for health outcomes.

## 5. Conclusions

HIIT training for 12 weeks may be an effective exercise intervention for improving glucose control, insulin sensitivity, cellular apoptosis, regulating mitochondrial biogenesis, and reducing oxidative stress biomarkers in individuals with T2DM mellitus. HIIT was suggested as an effective strategy for treating T2DM mellitus. However, further research is required to fully understand the mechanisms underlying these changes and their long-term implications for health outcomes.

## Figures and Tables

**Figure 1 medicina-59-01320-f001:**
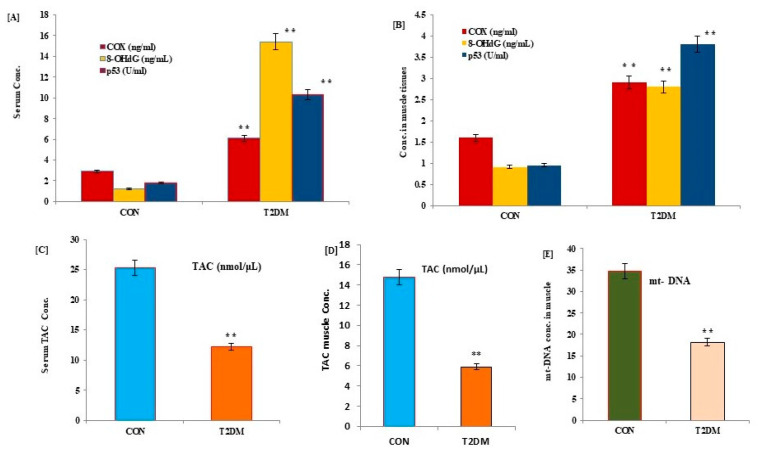
Changes in the levels of p53, COX, TAC, 8-OHdG, in serum and muscle tissues (**A**,**B**), TAC (**C**,**D**), and muscular mitochondrial DNA content (**E**) of control and T2DM participants. Data expressed as mean ± SD. ** *p* < 0.001 (pre vs. post) level values of control and diabetic patients. 8-OHdG: serum 8-hydroxyguanine; p53: tumor suppressor gene protein; TAC: total antioxidant capacity; COX: cytochrome c; CON: control; T2DM: type 2 diabetes mellitus; mt-DNA: mitochondrial DNA content.

**Figure 2 medicina-59-01320-f002:**
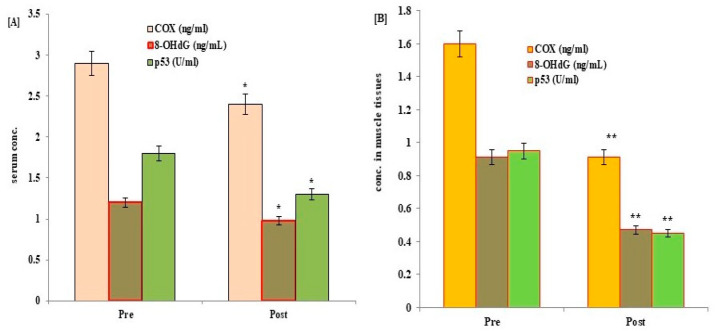
Changes in the levels of p53, COX and 8-OHdG in serum (**A**) and muscle tissues (**B**) of control healthy subjects following 12 weeks of a HIIT exercise program. Data expressed as mean ± SD. * *p* < 0.01, ** *p* < 0.001 (pre vs. post) level values of control subjects. 8-OHdG: serum 8-hydroxyguanine; p53: tumor suppressor gene protein; COX: cytochrome c oxidase; CON: control.

**Figure 3 medicina-59-01320-f003:**
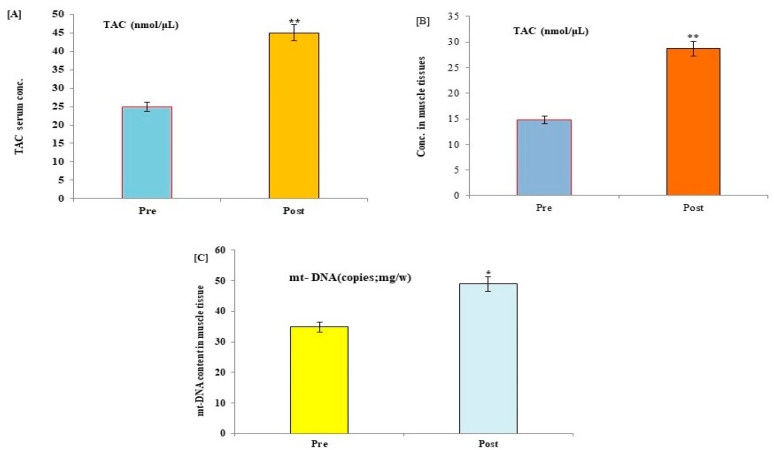
Changes in the levels of TAC and muscular mitochondrial DNA content in serum (**A**) and muscle tissues (**B**,**C**) of control healthy subjects following 12 weeks of a HIIT exercise program. Data expressed as mean ± SD. * *p* < 0.01, ** *p* < 0.001 (pre vs. post) level values of control subjects. TAC: total antioxidant capacity; mt-DNA: mitochondrial DNA content; CON: control.

**Figure 4 medicina-59-01320-f004:**
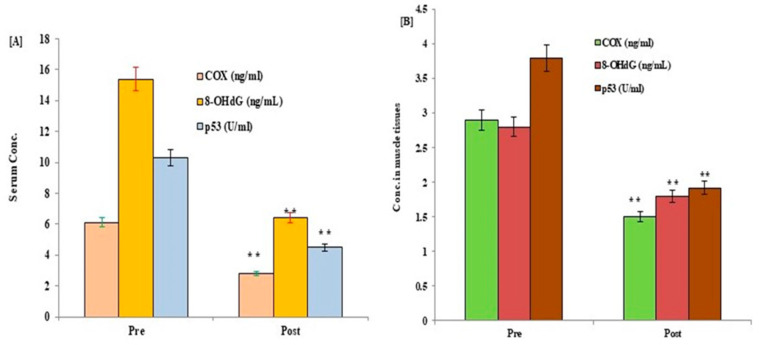
Changes in the levels of p53, COX, and 8-OHdG in serum (**A**) and muscle tissues (**B**) of type 2 DM participants following 12 weeks of a HIIT exercise program. Data expressed as mean ± SD. ** *p* < 0.001 (pre vs. post) level values of diabetic patients. 8-OHdG: serum 8-hydroxyguanine; COX: cytochrome c; p53: tumor suppressor gene protein; T2DM: type 2 diabetes mellitus.

**Figure 5 medicina-59-01320-f005:**
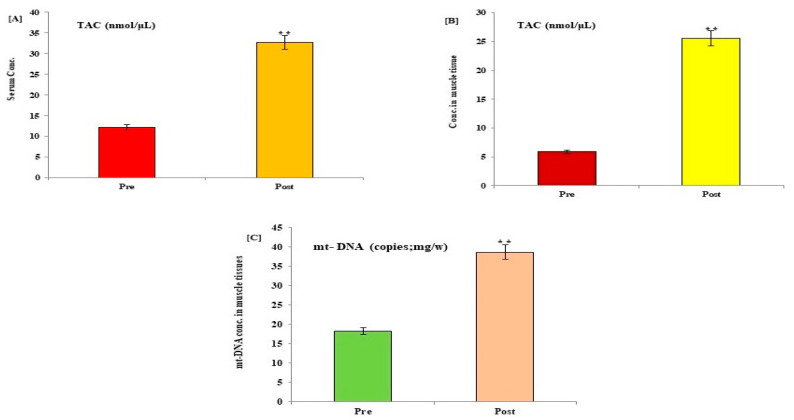
Changes in the levels of TAC and muscular mitochondrial DNA content in serum (**A**) and muscle tissues (**B**,**C**) of type 2 DM participants following 12 weeks of a HIIT exercise program. Data expressed as mean ± SD. ** *p* < 0.001 (pre vs. post) level values of diabetic patients. TAC: total antioxidant capacity; mt-DNA: mitochondrial DNA content; T2DM: type 2 diabetes mellitus.

**Table 1 medicina-59-01320-t001:** Improvements in adiposity and glycemic markers of control and type 2 DM subjects participated in HIIT exercise program for 12 weeks.

Parameters	CON (*n* = 20; Mean Age; 46.3 ± 2.8 Yrs.)		T2DM (*n* = 30; Mean Age: 46.1 ± 3.1 Yrs.)	
	Pre	Post	Pre	Post
BMI	24.5 ± 2.85	23.1 ± 1.6 *	31.8 ± 3.96	27.6 ± 2.7 **
Waist (cm)	98 ± 1.85	91.3 ± 1.1 *	156 ± 3.9	148.6 ± 2.4 *
Hips (cm)	115 ± 0.75	112 ± 0.81 *	67 ± 3.7	65.5 ± 2.7 *
WHR	0.85 ± 0.95	0.46 ± 0.89 *	2.33 ± 1.2	1.6 ± 0.97 **
Fitness score (VO_2_max; ml/kg × min)	25.8 ± 2.5	34.6 ± 4.6 *	21.3 ± 1.9	32.8 ± 2.8 **
Fasting glucose (mg/dL)	85.9 ±7.3	78.5 ± 2.8 *	165.2 ± 2.8	128.6 ± 3.7 **
Serum C-peptide (ng/mL)	3.95 ±1.7	4. 5 ± 3.9 *	2.8 ± 1.5	5.1 ± 1.5 **
HbA1c (%)	4.6 ± 0.45	3.2 ± 0.65 *	7.4 ± 1.6	5.2 ± 2.5 **
Fasting insulin (FI; μU/mL)	26.3 ± 7.9	32.8 ± 3.6 *	18.7 ± 5.8	35.9 ± 2.6 **
IR (mUmmol/L^2^)	5.3 ± 2.6	2.8 ± 1.9 *	12.9 ± 1.9	5.9 ± 3.4 **

Data expressed as mean ± SD, * *p* < 0.01. ** *p* < 0.001 (pre vs. post) level values of control and diabetic patients. BMI: body mass index; WHR: waist to hip ratio; IR: insulin resistance; VO_2_ max: maximal oxygen consumption; CON: control, T2DM: type 2 diabetes mellitus.

## Data Availability

All data generated or analyzed during this study are presented in the manuscript. Please contact the corresponding author for access to the data presented in this study.

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
