# Peer review of "High-Intensity Interval Training Improves Glycemic Control, Cellular Apoptosis, and Oxidative Stress of Type 2 Diabetic Patients"

_medicina, 2023, doi:10.3390/medicina59071320_

Round 1
Reviewer 1 Report
My impressions about the article are generally positive. However there are two minor points that the authors should answer:
1. For the purpose of the confirmation of the obtained results, the authors have to share or prove a link to the data used in this study. They provide processed results, but the core dataset must be also provided.
2. Please add more extended explanations related to function and impact of p53 and mitochondrial cytochrome c (COX) in the Introduction.
Author Response
Reviewer 1
Comments and Suggestions for Authors
My impressions about the article are generally positive. However, there are two minor points that the authors should answer:
Response: The study’s authors thank reviewer 1 for providing valuable comments to strengthen this manuscript to the best level for publication. We are committed to addressing the raised comments with our best capabilities.
Comment 1: For the purpose of the confirmation of the obtained results, the authors have to share or prove a link to the data used in this study. They provide processed results, but the core dataset must be also provided.
Response: The data will be uploaded to the journal’s site with the revised copy. Attached is the Excel file of the raw data of the parameters investigated in this study.
Comment 2: Please add more extended explanations related to function and impact of p53 and mitochondrial cytochrome c (COX) in the Introduction.
Response: A few paragraphs for more extensions and explanations about the function and roles of p53 and mitochondrial cytochrome c (COX) were added to the introduction part (Page 2; Lines 77-88).
Reviewer 2 Report
I have reviewed your article entitled "High-intensity interval training improves glycemic control, cellular apoptosis, and oxidative stress of diabetic patients" and would like to provide some comments and suggestions to improve the scientific clarity and consistency of the interpretation of your results.
1. Consider proofreading the manuscript for typographical errors and ensure consistent formatting of units and abbreviations.
2. Abstract:
- Consider specifying the type of diabetes being studied as "type-2 diabetes mellitus" throughout the abstract for consistency.
- Define HOMA-IR (homeostatic model assessment of insulin resistance) on first use to ensure readers unfamiliar with the term can understand its meaning.
- Instead of using abbreviations for some terms (e.g., FG for fasting glucose), consider using the full term for better clarity, especially for readers who may not be familiar with these abbreviations.
- Specify the units for each measurement mentioned, such as glucose levels (mg/dL or mmol/L), HbA1c (%), and 8-OHdG (ng/mL or μmol/L), to provide a better understanding for readers.
3. Please check the bullet numbering in each section carefully.
- Methods
- In the "Subjects" section, provide more specific information about the selection criteria for the healthy volunteers. Mention any exclusion criteria or specific characteristics that were considered when matching them with the diabetic patients. Additionally, include information about how the informed consent was obtained and any ethical considerations or approvals.
- Specify the rationale for choosing the age range of 45-60 years for the participants. Provide a brief explanation as to why this age group was relevant to the study objectives.
- In the "Skeletal muscle biopsy and blood samples" section, explain the reason for taking two resting biopsies—one week before performance testing and 72 hours after the last exercise training session. Elaborate on the specific muscle biopsy technique used and any considerations taken to ensure participant comfort and safety.
- When describing the quantification of genomic and mitochondrial DNA content, mention the specific method or assay used to estimate the copy numbers. Provide a brief overview of the protocol and its relevance to the study.
- Results
- In the sentence "Post-exercise data showed a significant reduction in the expression levels of p53, Cox, and 8-OHdG," it would be clearer to state the specific changes observed (e.g., decrease in expression levels) rather than using the term "reduction," which may be ambiguous.
- Use consistent terminology throughout the results section. For example, if "type 2DM or T2DM" is used as an abbreviation, ensure it is consistently used instead of alternating between "type 2 DM" and "diabetic cases."
- Proofread the text for any typographical errors or inconsistencies in formatting.
- Discussion
- Avoid using phrases like "Previous research studies showed positive and favorable outcomes" without providing specific references or evidence. Instead, explicitly state the findings of those studies and how they align with or differ from the current study.
- When discussing the effects of HIIT exercise training on mitochondrial parameters, such as p53, COX, and mtDNA content, provide more detailed explanations of the observed changes and their significance. Discuss the implications of these changes for mitochondrial function, oxidative stress, and cellular apoptosis.
- Consider discussing the practical implications of the study's findings. How might these results be applied in a clinical setting or for the development of exercise interventions for diabetic patients? Discuss the potential benefits and challenges of implementing HIIT exercise programs for diabetes management.
- Clearly state the limitations of the study and acknowledge any potential confounding factors or alternative explanations for the observed results. This will provide a balanced interpretation of the findings and demonstrate scientific rigor.
7. Conclusion
- Provide a concise summary of the key findings of the study without repeating information already discussed in the previous sections. Emphasize the most significant and novel aspects of the study.
- Avoid using vague statements like "The data showed that participation in HIIT program for 12 weeks significantly improves diabetes..." Instead, explicitly state the specific improvements observed in glycemic control, adiposity markers, mitochondrial parameters, and oxidative stress capacity.
- Highlight any novel contributions of the study to the existing body of literature. What new insights or knowledge does this study provide? How do the findings contribute to the scientific understanding of the effects of HIIT exercise training on diabetes?
- Consider mentioning any recommendations for future research based on the limitations or unanswered questions raised by the current study. This can help guide further investigations in the field
Author Response
Point-by-point response letter to reviewer comments
[Manuscript ID: medicina-2457265]
Reviewer 2
Comments and Suggestions for Authors: I have reviewed your article entitled "High-intensity interval training improves glycemic control, cellular apoptosis, and oxidative stress of diabetic patients" and would like to provide some comments and suggestions to improve the scientific clarity and consistency of the interpretation of your results.
Response: The study’s authors thank reviewer 1 for providing his valuable comments to improve the scientific clarity and consistency of interpreting the study’s results. We are committed to addressing the raised comments with our best capabilities.
Comment 1: Consider proofreading the manuscript for typographical errors and ensure consistent formatting of units and abbreviations.
Response: The entire study revised for typographical errors and formatting of units and abbreviations
Abstract
Comment 2: Consider specifying the type of diabetes being studied as "type-2 diabetes mellitus" throughout the abstract for consistency.
Response: In the abstract and throughout the revised manuscript, type 2 diabetes is specified as type-2 diabetes mellitus (T2DM)
Comment 3: Define HOMA-IR (homeostatic model assessment of insulin resistance) on first use to ensure readers unfamiliar with the term can understand its meaning.
Response: HOMA-IR checked and defined on first use (page 1; Line 27)
Comment 4: Instead of using abbreviations for some terms (e.g., FG for fasting glucose), consider using the full term for better clarity, especially for readers who may not be familiar with these abbreviations.
Response: all abbreviations revised and defined on first use throughout the revised manuscript.
Comment 5: Specify the units for each measurement mentioned, such as glucose levels (mg/dL or mmol/L), HbA1c (%), and 8-OHdG (ng/mL or μmol/L), to provide a better understanding for readers.
Response: All units were revised and the missing ones were added to their respective places (please see abstract, page 1; Lines 17-27)
Comment 6: Please check the bullet numbering in each section carefully.
Response: The manuscript has been revised, checked, and corrected for the bullet numbering accordingly (please see the method sections)
Methods
Comment 7: In the "Subjects" section, provide more specific information about the selection criteria for the healthy volunteers. Mention any exclusion criteria or specific characteristics that were considered when matching them with the diabetic patients. Additionally, include information about how the informed consent was obtained and any ethical considerations or approvals.
Response: All subjects were involved in this study following the assignment of the informed consent form. In addition, twenty healthy subjects from a population undergoing a standard annual physical examination and biological measurements for medical insurance were selected as controls. (please see Subjects section 2.1, Line 121-138)
Comment 8: Specify the rationale for choosing the age range of 45-60 years for the participants. Provide a brief explanation as to why this age group was relevant to the study objectives.
Response: The rationale of choosing the age range 45-60 years was clearly mentioned in the introduction part as per your valued comments (please see in the introduction, Lines 106-113).
Comment 9: In the "Skeletal muscle biopsy and blood samples" section, explain the reason for taking two resting biopsies—one week before performance testing and 72 hours after the last exercise training session. Elaborate on the specific muscle biopsy technique used and any considerations taken to ensure participant comfort and safety.
Response: The muscle biopsy was taken by a licensed medical general practitioner (GP) with experience in percutaneous needle biopsies. The biopsy was taken at baseline one week before exercise training to ensure the wound healing process and devoid of any inflammation before starting exercise. As reported previously, the second sample was taken 14 hours after the last exercise session. In addition, for all participants, the GP instructed the subjects to take the Band-Aid off 1 to 2 hours post-biopsy and allow the plasters to remove over time naturally. If there were any abnormalities with healing or severe pain post-biopsy, the subjects were instructed to communicate with the investigator or the GP, whom they help accordingly [ please see the methodology section, 2.3, Lines 155-168].
Comment 10: When describing the quantification of genomic and mitochondrial DNA content, mention the specific method or assay used to estimate the copy numbers. Provide a brief overview of the protocol and its relevance to the study.
Response: Real-time PCR analysis was performed to identify mitochondrial DNA content as previously reported. This section identifies mt-DNA in muscle tissues pre- and post-HITT exercise training for 12 weeks. The methodology was checked, revised, and corrected according to the previous studies. (please see methodology part, 2.4, Line 176-184).
Results
Comment 11: In the sentence "Post-exercise data showed a significant reduction in the expression levels of p53, Cox, and 8-OHdG," it would be clearer to state the specific changes observed (e.g., decrease in expression levels) rather than using the term "reduction," which may be ambiguous.
Response: The results were revised, and the terms increase, and decrease were used throughout the results (please see results).
Comment 12: Use consistent terminology throughout the results section. For example, if "type 2DM or T2DM" is used as an abbreviation, ensure it is consistently used instead of alternating between "type 2 DM" and "diabetic cases."
Response: T2DM is an abbreviation referring to subjects of diabetes mellitus in the entire manuscript.
Comment 13: Proofread the text for any typographical errors or inconsistencies in formatting.
Response: The text was revised and corrected for typographical errors or inconsistencies in formatting. (please see the entire sections of the results)
Discussion
Comment 14: Avoid using phrases like "Previous research studies showed positive and favorable outcomes" without providing specific references or evidence. Instead, explicitly state the findings of those studies and how they align with or differ from the current study.
Response: The discussion part was revised thoroughly, and most sections were reformatted as recommended (see discussion part).
Comment 15: When discussing the effects of HIIT exercise training on mitochondrial parameters, such as p53, COX, and mtDNA content, provide more detailed explanations of the observed changes and their significance. Discuss the implications of these changes for mitochondrial function, oxidative stress, and cellular apoptosis.
Response: HIIT has been shown to increase COX activity and expression in skeletal muscle and mt-DNA content in skeletal muscle, which may improve mitochondrial function and enhance cellular energy production. These changes may also have implications for cellular apoptosis. Mitochondria play a key role in regulating cellular apoptosis, and dysfunction of mitochondrial pathways can contribute to developing chronic diseases, including type 2 diabetes. HIIT may help to reduce the risk of cellular apoptosis and prevent the development of chronic diseases, especially in patients with T2DM mellitus, by improving mitochondrial function and reducing oxidative stress (please see discussion part, )
Comment 16: Consider discussing the practical implications of the study's findings. How might these results be applied in a clinical setting or for the development of exercise interventions for diabetic patients? Discuss the potential benefits and challenges of implementing HIIT exercise programs for diabetes management.
Response: When implementing HIIT exercise programs for diabetes management, the benefits of improved glucose control, cardiovascular fitness, and time efficiency may outweigh the potential risks for many individuals. In addition, HIIT can be performed using a variety of exercises and can be adapted to different fitness levels and physical limitations [97-100]. (please see discussion part, Lines 289-298).
Comment 17: Clearly state the limitations of the study and acknowledge any potential confounding factors or alternative explanations for the observed results. This will provide a balanced interpretation of the findings and demonstrate scientific rigor.
Response: Strengths and limitations of the study were added in separate sections following the discussion part (4.1 and 4.2).
Conclusion
Comment 18: Provide a concise summary of the key findings of the study without repeating information already discussed in the previous sections. Emphasize the most significant and novel aspects of the study.
Response: Revised and reformatted accordingly (see conclusion)
Comment 19: Avoid using vague statements like "The data showed that participation in HIIT program for 12 weeks significantly improves diabetes..." Instead, explicitly state the specific improvements observed in glycemic control, adiposity markers, mitochondrial parameters, and oxidative stress capacity.
Response: The discussion part was revised thoroughly, and most of the sections were reformatted as much as possible (please see discussion).
Comment 20: Highlight any novel contributions of the study to the existing body of literature. What new insights or knowledge does this study provide? How do the findings contribute to the scientific understanding of the effects of HIIT exercise training on diabetes?
Response: The novel contributions were mentioned in the study’s strength added at the end of the article (please see discussion part 4.1).
Comment 22: Consider mentioning any recommendations for future research based on the limitations or unanswered questions raised by the current study. This can help guide further investigations in the field.
Response: Further research based on larger cohort samples of patients with T2DM mellitus must fully understand the mechanisms underlying these changes and their long-term implications for health outcomes. (please see the discussion part, 4.2)
Reviewer 3 Report
This study aimed to determine the effects of a 12-week HIIT program on p53 expression, mitochondrial cytochrome c, and oxidative stress since little is known about the effects of High-intensity interval training (HIIT) on the p53 regulation mechanism and mitochondrial function in patients with T2DM.
I suggest some major revision:
- last and corresponding author name is missing: please check the name of all authors and their affiliation
- revise entirely the english language and the punctuation in all the manuscript because it is hardly comprehensible
- revise all the acronyms present in the manuscript text (also in the keywords): many acronyms were missing and many of them are wrong
- the control population was not age-related with the patients population and therefore the differences highlighted can not be totally ascribed to the presence of disease. Please expain why did you choose this kind of control. Moreover, the popoluation study is too little to have a strong statistics
- the licence code of the SPSS software is missing in the statistical analysis paragraph
- in the istograms, in all the black bar it is impossible to see the negative standard deviation: please change the color
- check the character dimension of the figure 4 legend and please render it uniform with the other legends
- enlarge the character dimension in all the figures, because it is hardly to read most of them
I suggest some major revions, expecially in the use of english language and of the punctuation that render the manuscript hardly comprehensive.
Author Response
Reviewer 3
Comments and Suggestions for Authors
This study aimed to determine the effects of a 12-week HIIT program on p53 expression, mitochondrial cytochrome c, and oxidative stress since little is known about the effects of High-intensity interval training (HIIT) on the p53 regulation mechanism and mitochondrial function in patients with T2DM. I suggest some major revision:
Response: The study’s authors thank the reviewer 2 for providing his valuable comments to strengthen this manuscript to the best level for publication. We are committed to addressing the raised comments with our best capabilities.
Comment 1: last and corresponding author name is missing: please check the name of all authors and their affiliations.
Response: All names and their affiliation were checked and corrected (please see it)
Comment 2: Revise entirely the English language and the punctuation in all the manuscript because it is hardly comprehensible.
Response: The entire manuscript was revised for English, checked for spelling mistakes, and revised for punctuation by a native language speaker from the same study area.
Comment 3: Revise all the acronyms present in the manuscript text (also in the keywords): many acronyms were missing, and many of them are wrong.
Response: All acronyms present in the study revised, checked, and missing were added accordingly (please, see in their respective places)
Comment 4: The control population was not age-related with the patient’s population and therefore the differences highlighted cannot be totally ascribed to the presence of disease. Please explain why did you choose this kind of control. Moreover, the population study is too little to have a strong statistic.
Response: To achieve a considerable HITT exercise influence, the control subjects were selected from a midle aged population undergoing a standard annual physical examination and biological measurements for medical insurance. The sample size calculation of the total poulation (n=50) using G ∗ Power program for Windows (version 3.1.9.7) achieves a power of 95% with an effect size of 0.92, Df = 45.74, critical t = 1.68, and noncentrality -α = 3.34. (please see 2.8. Sample calculation part of methodology).
Comment 5: The license code of the SPSS software is missing in the statistical analysis paragraph.
Response: All data levels set at p < 0.05 were reported as significant (SPSS statistical 13.0 software version for Windows; SPSS Inc., Chicago, IL, USA).
Comment 6: In the histograms, in all the black bar it is impossible to see the negative standard deviation: please change the color.
Response: All figures checked and reformatted again according to per valued comments (please see figures)
Comment 7: Check the character dimension of the Figure 4 legend, and please render it uniform with the other legends.
Response: All figures checked and reformatted again according to per valued comments (please see figures)
Comment 8: Enlarge the character dimension in all the figures, because it is hardly to read most of them.
Response: All figures reformatted and revised to be easy to read (please see figures)
Comments on the Quality of English Language.
Comment 9: I suggest some major revisions, especially in the use of English language and of the punctuation that render the manuscript hardly comprehensive.
Response: The entire manuscript was rechecked and revised for the English language, spelling mistakes, and punctuation by a native English speaker from the same study area.
Round 2
Reviewer 2 Report
I am grateful to you for the insightful clarification. The modifications that were made to the manuscript led to an increase in comprehension.
Reviewer 3 Report
The authors replied properly to all the comments made by the reviewer and modified the manuscript according to the reviewer's suggestions. The English language was drastically improved and now the manuscript has been strengthened to a better level for pubblication.